# Robust Contrastive Language-Image Pre-training against Data Poisoning and Backdoor Attacks

**Wenhan Yang**      **Jingdong Gao**      **Baharan Mirzasoleiman**
{hangeryang18, mxuan, baharan}@cs.ucla.edu
Computer Science Department, UCLA

## Abstract

Contrastive vision-language representation learning has achieved state-of-the-art performance for zero-shot classification, by learning from millions of image-caption pairs crawled from the internet. However, the massive data that powers large multimodal models such as CLIP, makes them extremely vulnerable to various types of targeted data poisoning and backdoor attacks. Despite this vulnerability, robust contrastive vision-language pre-training against such attacks has remained unaddressed. In this work, we propose RoCLIP, the first effective method for robust pre-training multimodal vision-language models against targeted data poisoning and backdoor attacks. RoCLIP effectively breaks the association between poisoned image-caption pairs by considering a relatively large and varying pool of random captions, and matching every image with the text that is most similar to it in the pool instead of its own caption, every few epochs.It also leverages image and text augmentations to further strengthen the defense and improve the performance of the model. Our extensive experiments show that RoCLIP renders state-of-the-art targeted data poisoning and backdoor attacks ineffective during pre-training CLIP models. In particular, RoCLIP decreases the success rate for targeted data poisoning attacks from 93.75% to 12.5% and that of backdoor attacks down to 0%, while improving the model's linear probe performance by 10% and maintains a similar zero shot performance compared to CLIP. By increasing the frequency of matching, RoCLIP is able to defend strong attacks, which add up to 1% poisoned examples to the data, and successfully maintain a low attack success rate of 12.5%, while trading off the performance on some tasks [1].

## 1 Introduction

Recent large-scale vision-language models pre-trained on millions of image-caption pairs crawled from the internet has gained an unprecedented success. Large-scale vision-language models such as CLIP (Radford et al., 2021) and ALIGN (Jia et al., 2021) are trained using a multimodal contrastive loss which pulls the representations of every image-caption pair together while pushing those of different pairs apart. In doing so, they can learn state-of-the-art image representations, without the need for a fixed set of predetermined labels to be specified at pte-training time. This enables zero-shot transfer of the model to downstream tasks, without requiring specialized output heads or dataset specific customization. Instead, natural language can be used to reference the learned visual concepts afterwards (Radford et al., 2021; Jia et al., 2021). Crucially, alleviating the need for expensive labeling of training examples enables scaling up the training data to millions of examples. However, the massive data that powers such large models also makes them extremely vulnerable to various types of targeted data poisoning and backdoor attacks (Carlini & Terzis, 2021; Yang et al., 2022).

---

[1] Code is available at `https://github.com/BigML-CS-UCLA/RoCLIP`

37th Conference on Neural Information Processing Systems (NeurIPS 2023).

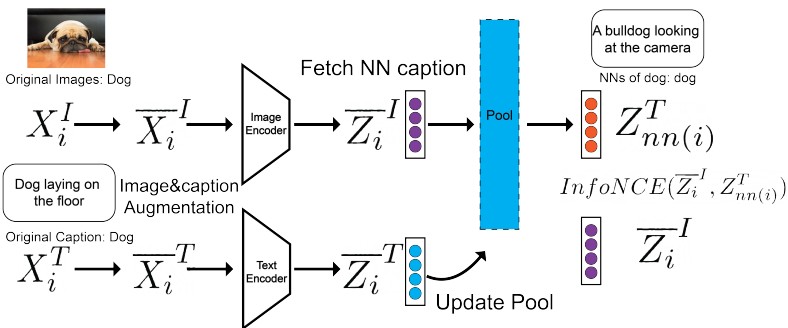

Figure 1: Illustration of RoCLIP. for defending CLIP during pre-training. (a) RoCLIP keeps a pool of random varying captions. During training, RoCLIP augments all images and captions. Then, it finds the most similar caption $z_{nn}^T(i)$ to an augmented image $\overline{z}_i^I$; and instead of matching the image $z_i^I$ and caption $z_i^T$, it matches every augmented image to its $z_{nn}^T(i)$.

Targeted data poisoning attacks on multimodal models add mismatched image-caption pairs to the pre-training data, to change the prediction of particular images at the test time. Similarly, backdoor attacks overlay a small patch on a subset of training data to cause the model to misclassify test images with the same patch. Notably, poisoning just 0.0001% of the pre-training examples can lead to success of a targeted poisoning attack. Similarly, poisoning 0.01% of pre-training examples can makes a backdoor attack successful (Carlini & Terzis, 2021). Compared to clean-label data poisoning and backdoor attacks in the supervised settings which require poisoning on average 1% of training data (Turner et al., 2018; Geiping et al., 2021), attacking multimodal contrastive models requires orders of magnitude fewer poisoned examples. Interestingly, the larger the model, the more vulnerable it is against adversarial attacks (Carlini & Terzis, 2021).

Despite this vulnerability, robust pre-training of multimodal vision-language models against targeted data poisoning and backdoor attacks has remained unaddressed. The recent work of Yang et al. (2022) studied poison identification during *fine-tuning* of CLIP, by using another CLIP model pre-trained on clean data to remove dissimilar image-caption pairs. Furthermore, Bansal et al. (2023) proposed CleanCLIP to eliminate the effect of backdoor attacks from a pre-trained CLIP model, by fine-tuning on a large subset of the clean pre-training data with in-modality contrastive loss on both vision and language modalities. However, such approaches are not applicable to pre-training, as we also confirm experimentally in Appendix 7.1.

In this work, we propose the first effective method, namely RoCLIP, for robust pre-training of multimodal vision-language models such as CLIP, against targeted data poisoning and backdoor attacks. Our approach is based on the following key observation: while the similarity between the image-caption pairs of clean examples increases rapidly during the training, similarity between poisoned image-caption pairs grows at a slower speed, early in training. As a result, poisoned images and captions are not close to the groups of similar images and captions in the representation space, during the initial training iterations. To break the association between poisoned image-caption pairs, our main idea is to keep a relatively large and varying pool of randomly selected captions. Then, we match every image with the text that is most similar to it in the pool instead of its original caption. This effectively prevents the attack by breaking the association between poisoned image-caption pairs, from early training epochs. We further strengthen our defense and improve the performance by leveraging image and text augmentations.

Our extensive experiments show that our method renders state-of-the-art targeted data poisoning and backdoor attacks ineffective during pre-training. In addition, our method leads to an increase of linear probe accuracy by up to 10% while having a zero shot performance on par with CLIP. By increasing the frequency of matching, RoCLIP is able to defend strong attacks, which add up to 1% poisoned examples to the data, and successfully maintain a low attack success rate of 12.5%, while trading off the performance on some tasks. We note that RoCLIP is the only effective defense method against state-of-the-art attacks that can efficiently scale to pre-training large-scale vision-language models such as CLIP.

## 2 Related Work

**Contrastive Representation Learning.** Contrastive learning was originally proposed for self supervised representation learning from unimodal data. Self-supervised contrastive learning works by maximizing agreement between differently augmented views of the same example and minimizing agreement between differently augmented views of different examples (Chen et al., 2020; Chen & He, 2021; He et al., 2020). Several works improved the performance of contrastive-learning on downstream tasks by imposing additional constraints to remove redundancy between components of the representation vectors and prevent collapse of the representations (Bardes et al., 2021; Zbontar et al., 2021), or using nearest-neighbor as positive pairs in the contrastive loss (Dwibedi et al., 2021; Van Gansbeke et al., 2021).

**Contrastive Language-Image Pretraining.** Multimodal vision-language models like CLIP (Radford et al., 2021) and ALIGN (Jia et al., 2021) are pre-trained on 400M/1B image-text pairs, by maximizing the agreement between image and text representations of every image-caption pairs and minimizing those of different pairs. A recent line of work aims at improving the data efficiency and quality of CLIP representations, by leveraging image and text augmentations. DeCLIP (Li et al., 2021) improves data-efficiency of CLIP by maximizing the similarity between two augmented image features using SimSiam (Chen & He, 2021), two augmented text features using Masked Language Modeling (MLM) (Devlin et al., 2018), and matching augmented image features with their augmented text pairs and other similar text features. SLIP (Mu et al., 2022) improves the performance by maximizing the agreement between two augmented image features using SimCLR (Chen et al., 2020), and matching the augmented image features with their text pair. CyCLIP (Goel et al., 2022) improves the representations by symmetrization of the similarity between the two mismatched image-text pairs, as well as the similarity between the image-image pair and the text-text pair. Finally, FILIP (Yao et al., 2021) uses transformer-based encoders for both modalities to learn more fine-grained features.

**Targeted Data Poisoning and Backdoor Attacks on CLIP.** Contrastive pretrained language-image models are extremely vulnerable to various types of targeted data poisoning and backdoor attacks (Carlini & Terzis, 2021). Targeted data poisoning attacks fool the model to misclassify a particular test example as an adversarial label. On the other hand, backdoor attacks overlay a small patch on a subset of training data, and cause the model to misclassify test images with the same patch. Despite this vulnerability, designing effective defense methods to protect the model from being poisoned during pre-training has remained unaddressed.

CLIP has been also shown to be vulnerable to data poisoning attacks during fine-tuning (Yang et al., 2022). To address this, Yang et al. (2022) proposed a pre-processing and a post-processing defense to defend against targeted data poisoning attacks during fine-tuning. The pre-processing requires a clean pre-trained CLIP to remove examples with low cosine similarity between image and their corresponding text representation. The post-processing fine-tunes the poisoned model on another clean dataset of the same scale as the fine-tuning data. Besides, Bansal et al. (2023) proposed a method to clean backdoor attacks from CLIP, by fine-tuning the model on a clean subset of the pre-training dataset with in-modality contrastive loss on both vision and language modalities. Such approaches are however not applicable to pre-training, as we also confirm experimentally in Appendix 7.1. In our work, we propose the first effective defense method that can protect the model from both targeted image attacks and backdoor attacks during pre-training.

## 3 Preliminary

### 3.1 Contrastive Language-Image Pre-training (CLIP)

CLIP is trained on millions of image-caption pairs scraped from the web. Formally, we consider a dataset $\mathcal{D} \subseteq \mathcal{I} \times \mathcal{T}$ consisting of pairs $(\boldsymbol{x}_j^I, \boldsymbol{x}_j^T)$ where $\boldsymbol{x}_j^I \in \mathcal{I}$ is a raw image and $\boldsymbol{x}_j^T \in \mathcal{T}$ is a text (caption of the image). The CLIP architecture consists of an image encoder $f_I : \mathcal{I} \rightarrow \mathbb{R}^d$ that encodes the raw image $x_i^I$ into an embedding vector $\tilde{\boldsymbol{z}}_i^I$, and a text encoder $f_T : \mathcal{T} \rightarrow \mathbb{R}^d$ that encodes the raw text $\boldsymbol{x}_i^T$ into an embedding vector $\tilde{\boldsymbol{z}}_i^T$ of the same dimension. Then projected image and text embeddings $\boldsymbol{z}_i^I, \boldsymbol{z}_i^T$ are obtained by passing the encoded image and text $\boldsymbol{z}_i^I, \boldsymbol{z}_j^T$ through their corresponding projection heads. The projected representations are normalized to have unit $\ell_2$-norm. Finally, the InfoNCE loss (Oord et al., 2018) is applied to pull the projected embeddings of every

image and its corresponding caption together while pushing apart the projected embeddings of the image from other captions in the same mini-batch. Formally, for a mini-batch of $N$ image-caption pairs $\{(\boldsymbol{x}_j^I, \boldsymbol{x}_j^T)\}_{j=1}^N$, and their projected embeddings $\{(\boldsymbol{z}_j^I, \boldsymbol{z}_j^T)\}_{j=1}^N$, the CLIP loss is defined as:

$$\mathcal{L}_{\text{CLIP}} = -\frac{1}{2N}\sum_{j=1}^N \log\left[\frac{\exp\left(\langle \boldsymbol{z}_j^I, \boldsymbol{z}_j^T\rangle /\tau\right)}{\sum_{k=1}^N \exp\left(\langle \boldsymbol{z}_j^I, \boldsymbol{z}_k^T\rangle /\tau\right)}\right] - \frac{1}{2N}\sum_{k=1}^N \log\left[\frac{\exp\left(\langle \boldsymbol{z}_k^I, \boldsymbol{z}_k^T\rangle /\tau\right)}{\sum_{j=1}^N \exp\left(\langle \boldsymbol{z}_j^I, \boldsymbol{z}_k^T\rangle /\tau\right)}\right],$$
(1)

where $\langle .,.\rangle$ represents the inner product, and $\tau$ is a trainable temperature parameter. The performance of the pre-trained CLIP model is evaluated via zero-shot and linear probe methods, as discussed below.

**Zero-shot classification.** Pre-trained Language-Image models such as CLIP enable zero-shot transfer of the model to downstream tasks, without the need for specialized output heads or dataset specific customization. To do so, the downstream labels is transformed into natural language captions using the provided engineered prompts templates, e.g. "A photo of a {label}". Then, the cosine similarity of the test image to each caption is computed, and the model predicts the label with the highest image-caption similarity.

**Linear probe.** For a labeled downstream image dataset, CLIP image representations can also be evaluated by training a linear classifier on the image representations obtained from the pre-trained CLIP image encoder and the downstream labels.

### 3.2 Targeted Poisoning and Backdoor Attacks

Let $\mathcal{D} = \{(\boldsymbol{x}_i^I, \boldsymbol{x}_i^T)\}_{i=1}^n$ be the set of all training examples. Poisoning attacks (Biggio et al., 2012) inject a small subset of poisoned examples $\mathcal{D}_p$ to the original training dataset $\mathcal{D}$, such that when the model is trained on the poisoned training data $\{\mathcal{D} \cup \mathcal{D}_p\}$, its prediction on particular test examples are changed to the adversarial label $y_{adv}$. At the same time, the poisoned model performs normally on other test examples. In this work, we consider both targeted poisoning and backdoor attacks as we discuss next.

**Targeted Image attacks.** In a targeted poisoning attack, the adversary aims to change the prediction of one particular test examples $\boldsymbol{x}_t^I$ to the adversarial label $y_{adv}$. Targeted poisoning attacks can be crafted following (Carlini & Terzis, 2021), by constructing a caption set $\mathcal{T}_{adv}$ of potential text descriptions related to the label $y_{adv}$, and making poisoned examples by assigning captions in $\mathcal{T}_{adv}$ to every target $\boldsymbol{x}_t^I$, i.e., $\mathcal{D}_p = \{(\boldsymbol{x}_t^I, \boldsymbol{x}_c^T) : \boldsymbol{x}_c^T \in \mathcal{T}_{adv}\}$. For constructing the caption set $\mathcal{T}_{adv}$, one can search the training dataset for all sequences that contain this label string, and use these sequences as the caption set. Alternatively, one can use the set of 80 different engineered prompt templates provided by CLIP for classification (Radford et al., 2021), and either use a subset or repeat them as necessary. The number of captions in $\mathcal{T}_{adv}$ determines the number of generated poisons per target. To evade automated cleaning algorithms (e.g., removing duplicated images), tiny Gaussian noise can be added to the images, or the captions can be modified by substituting or adding words, without degrading the attack success rate. A diverse caption set ensures that the image encoder is poisoned instead of the projection layers.

**Backdoor attacks.** In backdoor attacks, the adversary attaches a small trigger patch to the poisoned images and pair them with adversarial captions $\mathcal{T}_{adv}$ related to $y_{adv}$. In doing so, *all* the test images with the trigger patch will be misclassified as $y_{adv}$. In contrast to the targeted poisoning attack, instead of using a particular $\boldsymbol{x}_t^I$, the adversary poisons different images $\boldsymbol{x}_i^I \in \mathcal{I}$, by adding the trigger patch to them. Specifically, the poisoned set is defined as $\mathcal{D}_p = \{(\boldsymbol{x}_i^I \oplus \text{patch}, \boldsymbol{x}_c^T) : \boldsymbol{x}_c^T \in \mathcal{T}_{adv}, \boldsymbol{x}_i^I \in \mathcal{I}\}$. The caption set can be constructed in a similar manner to targeted poisoning attacks using captions found in the training data or engineered prompt templates.

In general, while injecting poisoned examples in curated datasets used for supervised learning might be difficult, such poisons can be easily injected in uncurated datasets used by large multimodal models. This makes such models highly vulnerable to adversarial attacks.

---

**Algorithm 1** Robust CLIP pre-training (RoCLIP)

---

1: **Input**: Image encoder $f_I$, text encoder $f_T$, caption pool $\mathcal{P} = \{z_i^T\}_{i=1}^P$ initialized with random captions, RoCLIP frequency $\mathcal{K}$

2: **for** epoch $= 1, \cdots, T$ **do**

3:     **for** every mini-batch of image-caption pairs: $\{(x_j^I, x_j^T)\}_{j=1}^N \in \mathcal{D}$ **do**

4:         Augment image and captions in the mini-batch: $\{(\bar{x}_j^I, \bar{x}_j^T)\}_{j=1}^N$

5:         **if** $T \% \mathcal{K} == 0$ **then**

6:             Pair every $\bar{x}_j^I$ with its nearest augmented caption in pool $z_{nn(j)}^T = \arg\min_{\overline{z_p}^T \in \mathcal{P}} \|\overline{z}_j^I - \overline{z}_p^T\|_2$

7:             Append the pool with captions in the mini-batch and discarding the oldest captions

8:             Train $f_I$ and $f_T$ with $\mathcal{L}_{\text{RoCLIP}}$ in Eq. 2

9:         **else**

10:          Train $f_I$ and $f_T$ with $\mathcal{L}_{\text{CLIP}}$ in Eq. 1

---

### 3.3 Threat Model

**Adversary Objective.** The primary objective of the adversary is to manipulate the output representations of CLIP, such that certain images are misclassified into adversarial categories instead of their true categories, while the other images are classified correctly.

**Adversary Capabilities.** We assume that the adversary has limited control over the pre-training data, and can inject a small number ($\leq 1\%$ of the dataset size) of poisoned examples into the training dataset. Adversary also has the knowledge of the model structure, the training algorithm, and the hyperparameter used by their victim, but they cannot modify the training process directly.

## 4 Robust Training of Multimodal Models against Targeted Data Poisoning and Backdoor Attacks

In this section, we first study the effect of data poisoning attacks on multimodal models. Then, we present our method for robust training of such models against data poisoning and backdoor attacks.

### 4.1 Effect of Data Poisoning Attacks on CLIP

We start by studying the effect of data poisoning attacks on multimodal models. By minimizing the CLIP contrastive loss in Eq. (1), the model changes such that every image representation $z_j^I$ moves towards its caption representation $z_j^T$ and gets far away from other (dissimilar) caption representations $z_k^T$. This makes corresponding categories of image and text (e.g. category of "Cat" or "Dog" in the image and text modality) to get closer to each other and get distant from other categories, during the pre-training. Crucially, as image-caption pairs belonging to a particular category are relatively similar to each other, their gradients have a large alignment with other examples in the same category. Therefore, categories of similar image-caption pairs insert a large cumulative gradient on the model and change the model quickly to get close to their pairs in the other modality.

On the other hand, image-caption pairs of poisoned examples are not similar to the other clean examples in the data. Hence, their gradient does not align well with the gradient of the clean examples. Therefore, by training on the poisoned pairs, the poisoned images and captions get far away from the other images and captions in the same category. That is, poisoned images become distant from other images in the same category, and adversarial captions become distance from other captions in the adversarial category. Next, we will rely on this observation to break the association between poisoned image-caption pairs.

### 4.2 Robust Training with RoCLIP

To prevent targeted data poisoning and backdoor attacks from being successful, we aim to break the association between the poisoned image-caption pairs. If this can be done, the poisoned image and caption representations do not get close enough to each other during training and the attack does not

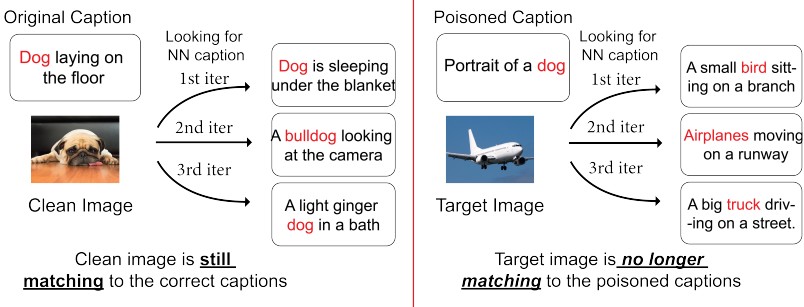

Clean image is **still matching** to the correct captions

Target image is **no longer matching** to the poisoned captions

Figure 3: Examples of matching the images in the clean or poisoned pairs during the training. The image in the clean pair can find semantically similar captions in the caption pool and still match to the correct captions, while the image in the poisoned pair will match to either random captions or captions corresponding to the original image.

succeed. To achieve this, we rely on our key observation in Sec. 4.1: poisoned images and captions are not close to groups of similar images and captions in the representation space, early in training. We leverage two techniques to break the association between poisoned image-caption pairs: (1) A large and varying pool of randomly selected captions; (2) Augmentations on both images and captions.

**The Pool.** As illustrated in Fig. 1, instead of matching every image with its corresponding caption, we match every image with the caption in the pool that is most similar to the image. Based on the observation from Sec. 4.1, poisoned captions are not close to other captions in the same category. Thus, our method prevents the poisoned images to be matched with captions from the adversarial category. In doing so, it breaks the data poisoning and backdoor attacks. Fig. 2 shows the effect of pool when training on a dataset with 15 poisoned images of a deer $x_t^I$ paired with adversarial truck captions $\{(x_t^I, x_c^T) : x_c^T \in \mathcal{T}_{\text{truck}}\}$. We see that while the majority of clean images are matched with captions from the same category as their original caption during training (blue line), poisoned images are matched with a caption from a different category than the adversarial label (orange line). This confirms our observation in Sec. 4.1, and shows the effectiveness of the pool. The pool may have a negative influence on the per-

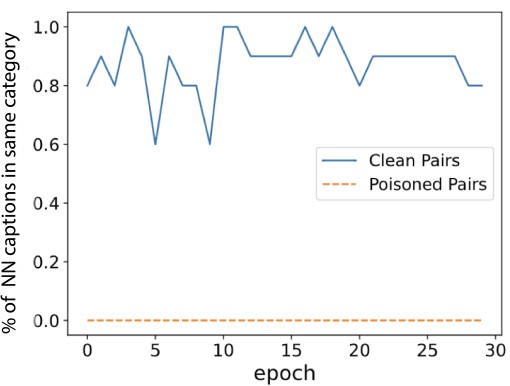

Figure 2: Training with ROCLIP. Majority of clean images are matched with captions from the same category as their original caption during training (blue line), while poisoned images are matched with a caption from a different category than the adversarial label (orange line).

formance, as it is unreliable especially early in training. To ensure a good model performance, we (1) select a relatively large pool size so that every clean image can find caption similar to its original caption; (2) train with our method every $\mathcal{K}$ epoch, and train with standard CLIP loss in the other epochs. Note that it is necessary to train on the poisoned pairs consistently to poison the model. With a smaller $\mathcal{K}$, the method is able to defend the model and disassociate the poisoned image-caption pairs. We select 2% of the total dataset size as our pool size and $\mathcal{K} = 3$ in our experiments. Fig. 3 illustrates an example of how ROCLIP disassociates poisoned pairs during the training.

**Augmentation.** To strengthen our defense, we use various image and text augmentations during the training. In particular, we use random image cropping, horizontal flipping, color jittering (Wu et al., 2018), grayscale conversion (Wu et al., 2018), and blurring (Chen et al., 2020) in our image augmentation policies. For the text augmentation, we use the EDA proposed by (Wei & Zou, 2019), which includes synonym replacement, random swap, and random deletion as its augmentation policies. The benefit of the augmentations are two-folded: First, similar to attacks on supervised learning (Borgnia et al., 2021), it help further reducing the attack success rates, but cannot fully defend the model, as we will confirm in our ablation study. Data augmentation is especially helpful during the

CLIP epochs to prevent matching poisoned image-caption pairs. Second, augmentations can further increase the method's performance, as shown in Table 3 and Table 7.

In summary, RoCLIP first samples a pool of $P$ caption representations $\mathcal{P} = \{z_i^T\}_{i=1}^P$ uniformly at random. During training, for every example $(x_j^I, x_j^T)$ in the mini-batch, we first augment its image and text with our augmentation policies, and then match its augmented image representation $\overline{z}_j^I$ with the augmented caption representation in the pool that is most similar to $\overline{z}_j^I$, i.e., $z_{nn(j)}^T = \arg\min_{\overline{z}_p^T \in \mathcal{P}} \|\overline{z}_j^I - \overline{z}_p^T\|_2$. Effectively, we form the positive image-caption representation pair $(\overline{z}_j^I, z_{nn(j)}^T)$ and use it instead of $(z_j^I, z_j^T)$. Similar to the CLIP loss, we obtain the negative pairs from the mini-batch. That is, for a mini-batch of $N$ image-caption pairs $\{(x_j^I, x_j^T)\}_{j=1}^N$, and their projected embeddings $\{(\overline{z}_j^I, \overline{z}_j^T)\}_{j=1}^N$, the loss is defined as:

$$\mathcal{L}_{\text{RoCLIP}} = \tag{2}$$

$$-\frac{1}{2N}\sum_{j=1}^N \log\left[\frac{\exp\left(\left\langle \overline{z}_j^I, z_{nn(j)}^T\right\rangle / \tau\right)}{\sum_{k=1}^N \exp\left(\left\langle \overline{z}_j^I, z_{nn(k)}^T\right\rangle / \tau\right)}\right] - \frac{1}{2N}\sum_{k=1}^N \log\left[\frac{\exp\left(\left\langle \overline{z}_k^I, z_{nn(k)}^T\right\rangle / \tau\right)}{\sum_{j=1}^N \exp\left(\left\langle \overline{z}_j^I, z_{nn(k)}^T\right\rangle / \tau\right)}\right].$$

For the pool $\mathcal{P}$, we consider a first-in-first-out queue, which is initialized with random caption representations. After training on every mini-batch, we update $\mathcal{P}$ by taking the caption representations of the $N$ examples in the mini-batch and concatenating them at the end of the queue. We discard the oldest $N$ elements from the queue, which equals to the training batch size.

The pseudocode of our method, RoCLIP, is illustrated in Alg. 1.

## 5 Experiments

In this section, we evaluate the effectiveness of RoCLIP in breaking targeted data poison and backdoor attacks while maintaining the model's performance. We evaluate our method for defending against various data poisoning and backdoor attacks, described in Sec. 3.2, during pre-training.

### 5.1 Training

We evaluate RoCLIP during pre-training on CLIP. For pre-training, we use an open-source implementation of CLIP as our model, with default ResNet-50 as the image encoder and Transformer as the text encoder. Each experiment is run with a batch size of 512 for 24 epochs, as the performance improvement afterwards is not significant, but more training makes the model more prone to being poisoned. We use Conceptual Captions 3M (CC3M) (Sharma et al., 2018) as our pre-training dataset. Due to limited computational resources, for pre-training we randomly sampled 1M image-caption pairs from CC3M as our training dataset. We assess our method on 10 downstream datasets introduced by (Kornblith et al., 2019), the detail of which can be found in Table 1.

Table 1: Details of downstream datasets.

| DATASET | CLASSES | TRAIN SIZE | TEST SIZE |
|---|---|---|---|
| CALTECH101 | 102 | 3,060 | 6,085 |
| CIFAR10 | 10 | 50,000 | 10,000 |
| CIFAR100 | 100 | 50,000 | 10,000 |
| DTD | 47 | 3,760 | 1,880 |
| FGVCAIRCRAFT | 100 | 6,667 | 3,333 |
| FLOWERS102 | 102 | 2,040 | 6,149 |
| FOOD101 | 101 | 75,750 | 25,250 |
| IMAGENET1K | 1000 | 50,000 | 50,000 |
| OXFORDIIITPET | 37 | 3,680 | 3,669 |
| STANFORDCARS | 196 | 8,144 | 8,041 |
| CONC. CAPT. (1M) | - | 1,000,000 | - |

### 5.2 Attack Methods

We consider targeted image attacks and backdoor attacks, discussed in Sec. 3.2.

**Targeted Image Attacks.** In our pre-training experiment, we choose a random target image $x_t$ from the conceptual captions validation set, and then choose a random target class from the ImageNet test set to generate a set of $|\mathcal{T}_{adv}|$ adversarial captions. Note that, (Carlini & Terzis, 2021) pre-trained 32 CLIP models and measured the attack success rate as the fraction of positioned models. This requires 3200 GPU hours. To reduce the computation, we poison 16 different random images by generating $|\mathcal{T}_{adv}|$ adversarial captions for each image? related to a label selected at random from ImageNet.

Table 3: Linear probe and zero-shot top 1 accuracy. RoCLIP improves the performance of the model by up to 10% for linear probe and obtains a similar zero-shot performance compared with CLIP.

| Method | Task | F102 | Fd101 | I1K | Pet | Cars | Cal101 | C10 | C100 | DTD | Air. |
|--------|------|------|-------|-----|-----|------|--------|-----|------|-----|------|
| RoCLIP | 0-shot | 0.83 | 6.34 | 6.63 | 3.68 | 0.72 | 30.38 | 30.14 | **9.52** | 3.56 | **1.11** |
|        | lin-prb | 99.22 | **54.05** | **24.09** | **52.36** | **20.35** | **72.15** | **78.99** | **57.82** | **55.21** | **32.55** |
| CLIP | 0-shot | 1.0 | 7.1 | 9.6 | 3.4 | 0.8 | 34.9 | 34.9 | 7.3 | 3.7 | 0.8 |
|      | lin-prb | 99.5 | 44.9 | 22.2 | 48.2 | 12.9 | 70.4 | 70.5 | 45.8 | 48.2 | 24.9 |

Then, we report the attack success rate as the fraction of images that are classified as the adversarial label, in a single pre-training run. In doing so, the attack success rate will be at least as high as (Carlini & Terzis, 2021). In addition, note that attacking our smaller dataset of 1M examples also results in a higher attack success rate compared to that of 3M used by (Carlini & Terzis, 2021). We will first evaluate the effectiveness of RoCLIP in defending a moderate amount of poisons ($|\mathcal{T}_{adv}| = 50$) without any performance loss. Then, we will examine if RoCLIP can defend very strong data poisoning attacks ($|\mathcal{T}_{adv}| = 100 - 10,000$) successfully.

**Backdoor Attacks.** We use the public Hidden Trigger Backdoor Attacks (HTBA) patches (Saha et al., 2020), that are square triggers generated by drawing a random 4×4 matrix of colors and resizing it to the desired patch size using bilinear interpolation. We use a resized 16×16 patch and put it consistently on the left top corner of the image. In our pre-training experiments, we randomly select 150 images from the CC3M validation dataset and pair them with adversarial captions related to a random target class from ImageNet. To evaluate the effectiveness of the backdoor attacks, we select 300 random images from the ImageNet validation set and patch them in the left top corner to measure the attack success rate.

## 5.3 RoCLIP Robustly Pre-trains CLIP

First, we evaluate the effectiveness of our method, RoCLIP, against targeted poisoning and backdoor attacks during the pretraining phase. We present our result in Table 2, where attack success rate shows the fraction of poisoned or backdoored images successfully classified as the desired label. We see that without any defense, 93.75% of the total poisoned images are

Table 2: RoCLIP defense performance.

| Model | Attack | Success Rate |
|-------|--------|--------------|
| CLIP | Target Img | 93.75% |
|      | Backdoor | 78% |
| RoCLIP | Target Img | **12.5%** |
|        | Backdoor | **0%** |

classified to the desired target class, and 78% of the backdoored images are classified as the target class. On the other hand, RoCLIP is able to fully defend the attack and reduce the attack success rate to 0% for the backdoor attacks and as low as 12.5 % for the targeted poisoning attacks. This clearly confirms the effectiveness of RoCLIP in breaking various types of data poisoning and backdoor attacks on CLIP during pre-training.

### 5.3.1 RoCLIP does not Harm the Performance

Next, we evaluate if RoCLIP negatively impacts the model performance. We assess the performance of RoCLIP on a variety of datasets introduced by (Kornblith et al., 2019), the detail of which can be found in Table 1. We evaluate RoCLIP with both zero-shot and linear-probe methods and present the result in Table 3. It can be seen that RoCLIP does not harm the overall model performance. In contrast, RoCLIP effectively improves the linear-probe classification performance across all ten datasets by up to 10% and has an on-par zero-shot performance with CLIP. As we will show in our ablation study, data augmentation contributes the most to RoCLIP's performance boost. Nevertheless, even without data augmentation, RoCLIP achieves a similar performance with CLIP.

### 5.3.2 RoCLIP against Very Strong Attacks

Next, we consider much stronger poisoning attacks against CLIP during pre-training. Note that, in general a much lower poison rate is enough to successfully poison multimodal models (0.0001%),

Table 4: RoCLIP against very strong data poisoning attacks RoCLIP can defend the model with some performance tradeoff. Linear probe and zero-shot performance is reported on CIFAR-10 (C10), CIFAR-100 (C100), Food101 (FD) and Caltech101 (Cal).

| | Poison Rate | | | | Zero-Shot | | | | Linear-Probe | | | |
|---|---|---|---|---|---|---|---|---|---|---|---|---|
| | 0.01% | 0.0512% | 0.5% | 1.0% | C10 | C100 | FD | Cal | C10 | C100 | FD | Cal |
| **RoCLIP** | **0%** | **0%** | **0%** | **12.5%** | 21.3 | 7.3 | 3.4 | 17.0 | 69.7 | 46.6 | 45.6 | 59.8 |
| CLIP | 100% | 100% | 100% | 100% | 35.0 | 7.3 | 7.1 | 34.9 | 70.5 | 45.8 | 44.6 | 70.4 |

Table 5: Linear-Probe (LP) performance, Poison Success Rate (PSR), and Backdoor Success Rate (BSR) with various pool sizes used in RoCLIP.

| Pool Size | 256 (0.2%) | 2048 (2%) | 21845 (20%) | 0 (CLIP w/ Aug) | 0 (CLIP) |
|---|---|---|---|---|---|
| Memory(MB) | 39449 | 39627 | 39505 | 39342 | 37459 |
| Time(s) | 538.31 | 538.69 | 539.47 | 536.96 | 506.02 |
| C10 LP | 64.5% | 67.1% | 67.9% | **72.9%** | 62.1% |
| C100 LP | 34.9% | 39.6% | 41.1% | **48.4%** | 37.4% |
| PSR | **0%** | **0%** | **0%** | 12.5% | 75% |
| BSR | **0%** | **0%** | **0%** | 0.33% | 37.5% |

compared to supervised learning models (1%), and hence the highest poison rate considered in (Carlini & Terzis, 2021) is 0.017%. Considering the very large size of the pretraining data for multimodal models, this small poison rate translates to a considerable "poison number" that needs to be generated by the adversary. Nevertheless, we evaluate the effectiveness of RoCLIP against a very high poison rate of up to 1% of the dataset size for targeted image attacks. To defend the attacks effectively, we change the frequency from $\mathcal{K} = 3$ to $\mathcal{K} = 2$. Table 4 shows that RoCLIP with $\mathcal{K} = 2$ can effectively defend very stronger attacks, while trading off the performance on some tasks.

### 5.3.3 Comparison to Fine-tuning Defense Baselines

Finally, we compare the performance of RoCLIP with existing methods proposed to defend CLIP during fine-tuning (Bansal et al., 2023; Yang et al., 2022). Due to page limit, we leave the discussions in the Appendix 7.1. Our results confirm that such methods are indeed highly ineffective in defending CLIP during pre-training.

### 5.4 Ablation Studies

In this section, we conduct further experiments on different components of RoCLIP to evaluate its effects on defense and downstream performance. All experiments are conducted on 100K data randomly sampled from CC3M, with 15 poisoned pairs and 90 backdoored pairs. We train the model with a batch size of 256 for 30 epochs.

### 5.4.1 RoCLIP against Other Backdoor Attacks

In Sec. 5.3, we explored RoCLIP's effectiveness in defending CLIP against patched backdoor attacks. Here, we evaluate RoCLIP against three additional backdoor attacks baselines, Blended, WaNet, and label-consistent attacks, used in (Bansal et al., 2023). We consider 90 pairs of backdoored image-caption pairs for Blended triggers, 256 pairs for WaNet triggers, and 512 pairs of label-consistent pairs. Table 6 shows that RoCLIP is able to reduce the backdoor success rate to nearly 0% for all three backdoor attacks.

Table 6: RoCLIP against different backdoors.

| Model | Attack | Backdoor Rate | BSR |
|---|---|---|---|
| CLIP | Blended | 0.09% | 28.00% |
| | WaNet | 0.256% | 43.40% |
| | Label-Consis | 0.512% | 97.30% |
| RoCLIP | Blended | 0.09% | **0.33%** |
| | WaNet | 0.256% | **0.67%** |
| | Label-Consis | 0.512% | **0.00%** |

Table 7: Effect of pool and data augmentation on RoCLIP.

| Method | P+A | Pool (P) | Aug (A) | CLIP |
|--------|-----|----------|---------|------|
| C10 LP | **67.1%** | 60.1% | 72.9% | 62.1% |
| C100 LP | **39.6%** | 36.5% | 48.4% | 37.4% |
| PSR | **0%** | 0% | 12.5% | 75% |
| BSR | **0%** | 0.67% | 0.33% | 42.3% |

Figure 4: RoCLIP against more challenging poisoned pairs.

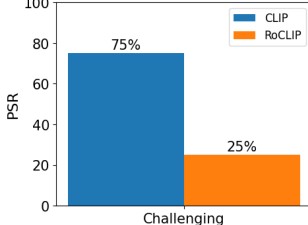

### 5.4.2 RoCLIP with Different Pool Size

Next, we analyze the computation overhead of RoCLIP, and the effect of pool size on our method. We apply RoCLIP with pool size of $0.2\%$, $2\%$ and $20\%$ of the pre-training dataset size. As shown in Table 5, RoCLIP is very lightweight, and has a negligible memory overhead. The overall memory usage of RoCLIP is similar to that of CLIP. In addition, the run-time overhead is also not significant, and is mostly caused by data augmentation. Second, the size of the pool does not have a significant influence on the defense. With different pool sizes, our method is able to defend the both the targeted poisoning and the backdoor attacks. With a larger pool size, however, the downstream performance of the model increases since it is more likely for the images to find fitting captions in a larger pool.

### 5.4.3 RoCLIP with Augmentations

Finally, we analyze the effect of data augmentations on our method. Table 7 shows that data augmentation improves the downstream performance. Note that even without data augmentation, RoCLIP obtains a similar linear probe accuracy with CLIP. Data augmentation further improves the performance of RoCLIP by $5\%$ on CIFAR-10 and $2\%$ on CIFAR-100. In addition, data augmentation significantly improves the defense and reduces the poison success rate from $75\%$ to $12.5\%$ and backdoor success rate from $42.3\%$ to $0.33\%$. Using both the pool and data augmentation, RoCLIP is able to reduce the success rate of backdoor and target image attacks to $0\%$ with a superior downstream performance.

### 5.4.4 RoCLIP against Adaptive Attacks

One potential adaptive attack against RoCLIP is to use visually similar categories when selecting the poisoned pairs, e.g., dog and cat. Note that, this strongly limits the choice of attackers' target images as well as adversarial captions. To evaluate the effectiveness of RoCLIP against such challenging poisoned pairs, we include 15 poisoned image-caption pairs from 8 visually-similar-categories like dog and cat, football and basketball (*c.f.* Appendix 7.2). Fig. 4 shows the effectiveness of RoCLIP.

## 6 Conclusion

We proposed RoCLIP, an effective method for robust training multimodal vision-language models such as CLIP against data poisoning and backdoor attacks. RoCLIP utilizes a caption pool as well as data augmentation to break the associations between the poisoned image and caption pairs, thus effectively defending the models. Through extensive experiments, we demonstrated that our proposed method drops the success rate of targeted data poisoning and backdoor attacks to 12.5% and 0% respectively. At the same time, it improves the model's performance by up to 10% on linear-probe and obtained a comparable zero shot performance to the baseline CLIP model. By increasing the frequency of matching, RoCLIP is able to defend very strong attacks with 1% poison rate and maintain a low attack success rate of 12.5%, while trading off the performance on some tasks.

**Limitations.** RoCLIP improves the robustness of CLIP pre-training against targeted data poisoning and backdoor attacks. Considering the demand for such models, our work makes a valuable step towards safe machine learning. RoCLIP can defend against a moderate number of poisons with no performance loss and against a high number of poisons with some performance tradeoff. Developing stronger defense methods without performance loss is an interesting direction for future work.

**Acknowledgment.** This research was supported by the National Science Foundation CAREER Award 2146492 and Cisco Systems.

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

# 7 Appendix

## 7.1 Adapted fine-tuning defense baselines

Here, we apply the methods proposed in (Yang et al., 2022; Bansal et al., 2023) to defend CLIP against data poisoning attacks or eliminate backdoor attacks from a pre-trained model during fine-tuning. We apply these methods during pre-training CLIP and show that they are indeed highly ineffective in protecting the model.

**Pre-processing CLIP** First, we show that pre-processing CLIP is not effective during pretraining. We conduct our experiment from the 1M sub-dataset randomly-sampled from CC3M. Yang et al. (2022) proposed using a pretrained model to calculate and compare cosine similarity between poisoned image-caption pairs and clean image-caption pairs. Then, it finds a threshold to separate the majority of poisoned pairs from clean pairs, and discard all the pairs with lower similarity than the threshold. We applied the above idea to pretraining, in a cheating experiment. That is, we consider the poisoning ratios of 0.005%-0.5%, and calculate the cosine similarity between the known generated poisoned image-caption pairs (note that in practice the poisons are not known) at epoch 1. We chose epoch 1, as the separation between cosine similarity of poisoned and clean image caption paris is largest at epoch 1 (see Figure 2). Then we set the threshold such that a certain fraction of the poisoned pairs are discarded, and filter examples with cosine similarity lower than this threshold.

Table 8: Ratio of clean data pairs discarded by Pre-processing CLIP when it discard a fixed ratio of poisoned data pairs.

| Discard/Poison % | 0.005% | 0.01% | 0.1% | 0.5% |
|---|---|---|---|---|
| 75% | 15.7% (*) | 44.3% | 98.3% | 95.8% |
| 85% | 23.3% (*) | 62.3% | 98.9% | 96.3% |
| 95% | 38.1% | 87.7% | 99.55% | 97.1% |

Tab. 8 shows the fraction of clean pairs that are discarded using different thresholds. We see that regardless of the threshold, a very large fraction of clean pairs are discarded. We trained and evaluated the two models when less than 30% of the clean data was discarded, marked with (*), and show the results in Tab. 9. Both models with smallest amount of clean pairs removed were still poisoned. With 75% and 85% of the poison discarded, the targeted data poisoning attacks still have 53.84% poison success rates. This confirms the ineffectiveness of the above method during pretraining. On the other hand, ROCLIP is able to drop the success rate to 0% (using $\mathcal{K} = 2$) and 12.5% (using $\mathcal{K} = 3$), and provides a much better trade-off between the attack success rate and model's performance and can defend a much higher poison number up to 1%.

Table 9: Defense and downstream performance of the Pre-processing CLIP compared to ROCLIP.

| Model | $\mathcal{K}$ | Poison % | Discard % | PSR | LP C10 | LP C100 | ZS C10 | ZS C100 |
|---|---|---|---|---|---|---|---|---|
| CLIP | - | 0.005% | - | 93.75% | 70.5 | 45.8 | 34.9 | 7.2 |
| Pre-proc. CLIP (*) | - | 0.005% | 75% | 58.3 | 70.2 | 45.6 | 27.9 | 7.3 |
| Pre-proc. CLIP (*) | - | 0.005% | 85% | 58.3 | 69.9 | 45.5 | 24.0 | 6.5 |
| ROCLIP | 3 | 0.005% | - | 12.5% | 78.9 | 57.8 | 30.1 | 9.5 |
| ROCLIP | 2 | 0.005% | - | 0% | 69.7 | 46.6 | 21.3 | 7.27 |
| ROCLIP | 2 | 0.1% | - | 0% | 69.7 | 46.6 | 21.3 | 7.27 |
| ROCLIP | 2 | 0.5% | - | 0% | 69.7 | 46.6 | 21.3 | 7.27 |
| ROCLIP | 2 | 1% | - | 12.5% | 69.7 | 46.6 | 21.3 | 7.27 |

We note that, the key factor for the success of ROCLIP is that poisoned images and captions are not close to the other images and captions in the same category. For example, a poisoned dog image that is captioned adversarially as a deer, remains far away from the rest of dog images in the representation space, during the initial training epochs. Hence, nearest-neighbor matching done by ROCLIP can effectively break the association of poisoned image caption pairs.

**CleanCLIP** Next, we show that CleanCLIP is not effective during pretraining. We conduct our experiment from the 100K sub-dataset randomly-sampled from CC3M. (Yang et al., 2022) proposed fine-tuning on a clean dataset (of the same size as pretraining) with CLIP loss, and (Bansal et al., 2023) proposed fine-tuning on a clean subset of size 100K of the original training dataset (same size as the pretraining data we use in this experiment) with CLIP loss+in-modality contrastive loss on both image and text modalities. Since a clean dataset is not available during pretraining, here we use the CleanCLIP loss (CLIP loss + in-modality contrastive loss) to pretrain CLIP on the (poisoned)

Table 10: Defense and downstream performance of CleanCLIP compared to ROCLIP.

| Model | Poison % | Backdoor % | PSR | BSR | LP C10 | LP C100 |
|---|---|---|---|---|---|---|
| ROCLIP | 0.015% | 0.15% | 0% | 0% | 67.1 | 39.6 |
| CleanCLIP | 0.015% | 0.15% | 25% | 49% | 67.7 | 43.2 |
| CLIP | 0.015% | 0.15% | 75% | 49% | 62.1 | 37.4 |

data. We conduct our experiment on 100K data, with 150 pairs of backdoored image-captions pairs and 15 targeted poisoned pairs. The following table shows that while the in-modality loss enables CleanCLIP to achieve a good performance, CleanCLIP cannot successfully defend against targeted data poisoning and backdoor attacks during pretraining. On the other hand, ROCLIP effectively drops the attack success rate to 0% for both targeted poisoning and backdoor attacks. This further confirms the effectiveness of ROCLIP.

## 7.2 Challenging poisons

For the visually challenging poisons, we consider 8 pairs of categories: cat-dog; shark-goldfish; snake-alligator; bee-butterfly; deer-buffalo; soccer-basketball; hammer-screwdriver; train-minivan.

