# OpenReview forum: "Robust Contrastive Language-Image Pretraining against Data Poisoning and Backdoor Attacks"
_NeurIPS.cc/2023/Conference — NeurIPS 2023 poster_

### Official Review · Reviewer_e25B · 2023-06-24

**Soundness:** 3 good
**Presentation:** 3 good
**Contribution:** 3 good
**Rating:** 6
**Confidence:** 4

**Summary:**

This paper considers a robust CLIP (RoCLIP) pretraining procedure against targeted and backdoor data poisoning attacks. Specifically, instead of minimizing the InfoNCE loss on the exact image-text pairs, RoCLIP considers matching an image with a caption that has the largest similarity, choosing from several candidates (e.g., a memory bank of previous text representations). Consequently, this may break the poisoned image-text pairs and reduce the chance of being poisoned.

**Strengths:**

The power of CLIP largely relies on formulating one-to-one image-text pairs as positive samples. Thus when this association breaks (e.g., through data poisoning), CLIP fails to be robust. This paper addresses this problem using a simple but seems effective idea: if one can actually choose positive pairs among a memory bank (ranked according to similarity), the large reliance on exact matching can be alleviated, thus CLIP becomes robust to false label attacks.

Overall I had doubts regarding several aspects of this approach (e.g., Does it hurt the performance of CLIP? Does it hurt the training efficiency? ) when reading the paper, but the experiments are well executed, and I am convinced this approach could be useful to boost the robustness of CLIP.

**Weaknesses:**

I have several remaining doubts and questions which I will summarize below. I hope to discuss them with the authors more during the rebuttal session.

**1. Early in training**

The authors illustrate in Section 4.1 and Figure 2 (one minor thing, Figure 2 is below Figure 3) that "poisoned images and captions are not close to groups of similar images and captions in the representation space, *early in training*" and use it as the main motivation of this paper. Why is *early in training* so important here and throughout the paper?  Seems that without this observation your motivation and proposed approach still hold (your algorithm is not changed dynamically during training).

**2. The caption of NN**

What do you mean by NN caption? What does NN suggest?

**3. The power of the encoders**

To rank the similarity between $z_{j}^{I}$ and $z_{p}^T$ properly, it seems that you assume that you have the perfect encoder already. How do you ensure this in the early stage of training? Is it possible that the algorithm actually chooses the wrong caption?

**4. The number of poisoned samples**

To me, the success of the algorithm heavily relies on the condition that clean samples overwhelm the poisoned samples such that the false captions do not dominate in the pool. Thus, when there are more poisoned samples in the training set, is your algorithm still effective against data poisoning attacks?







**Questions:**

See the section above.

**Limitations:**

Limitations are well addressed.

---

> ### Author Rebuttal · Authors · 2023-08-10
>
> We thank the reviewer for recognizing the effectiveness, and simplicity of our method, and the proficiency of our experiments.
>
> (1) For the reason of using "poisoned images and captions are not close to groups of similar images and captions in the representation space, early in training" as our main motivation:
>
> As correctly mentioned by the reviewer, our method relies on the key observation that poisoned images and captions are not close to groups of similar images and captions in the representation space, early in training (compare green and blue line before epoch 5 in Fig 2).
>
> As a result, when RoCLIP pairs an image with its nearest neighbor caption early in training, the (poisoned) image is not associated with a caption from the poisoned category (since the image is not close to the poisoned caption category yet). In doing so, RoCLIP can prevent associating the poisoned image-caption pairs from the beginning and subsequently during the entire training.
>
> In contrast, as clean image-caption pairs are close to the other images and captions in their category from the beginning of training, by associating a clean image with its nearest neighbor caption, the image is associated with a caption from its original category (not an irrelevant caption). Indeed, it is crucial to start RoCLIP early in training when poisoned image-captions are not yet close. This is because every time an image-caption pair is trained on with the CLIP loss, they are pulled together and their cosine similarity increases.
>
> RoCLIP prevents the CLIP loss to pull poisoned images and captions closer (see orange line in Fig 2). Otherwise, training with the CLIP loss on the poisoned images and captions quickly pulls them together and the attack becomes successful after a few epochs. At this point, the nearest neighbor of a poisoned image is from a poisoned category, and RoCLIP won’t be effective anymore.
>
> While the reviewer is right that we do not change our method dynamically during the training, it is important to note that the cosine similarity of image-caption pairs that are trained on increases during the training. As a result, images are paired with different captions at different epochs.
>
> (2) The NN denotes the nearest neighbor. We will clarify that in our revised version, thank you!
>
> (3) Assumption of having perfect encoder:
>
> We do not assume that we have the perfect encoder at an early stage of the training (or during the training). In fact we use the representations of the model being trained. Hence, the algorithm may choose the wrong captions in the first few epochs. To break the attacks while preserving the performance of the model, we apply NN matching every K=3 epochs, as discussed in lines 204.
>
> (4)  To illustrate that our method works against attacks with higher poison ratios:
>
> We increased the adversarial captions to {100, 256, 512}. Note that this  targeted poison ratio (0.000512) is higher than the max poison ratio (0.00017) considered in Carlini et al. As shown from the table below, our method is able to defend the model. Note that, to defend more poisons, we used RoCLIP with an increased K=2 (vs K=3 in the paper), where K is the frequency of applying the NN loss. This results in a stronger defense against attacks at the expense of slightly compromising some of the downstream performances. Specifically, there is a drop in the zero-shot performance on CIFAR10, but the linear-probe performance and the zero-shot performances on other datasets are still on par with CLIP.
>
>
> | Model  | # poison    | PSR      | LP C10 | LP C100 | ZS C10 | ZS C100 |
> |--------|-------------|----------|--------|---------|--------|---------|
> | RoCLIP | 100/256/512 | 0%/0%/0% | 69.7%  | 46.6%   | 21.3%  | 7.27%   |
> | CLIP   | -           | -        | 70.5%  | 45.8%   | 30.14% | 7.29%   |

---

> > ### Comment · Reviewer_e25B · 2023-08-10
> > **Thank you for your rebuttal**
> >
> > My concerns are well addressed, and I have raised my score accordingly.

---

> > > ### Author Response · Authors · 2023-08-18
> > >
> > > We thank the reviewer for the positive feedback and increasing your score.
> > >
> > > We’d also like to mention our new experimental evaluations, confirming that:
> > >
> > > (1) RoCLIP can effectively defend much higher poison ratios (0.5-1%): https://openreview.net/forum?id=ONwL9ucoYG&noteId=WYaDLJFW0j
> > >
> > > (2) RoCLIP can successfully defend a new backdoor attack: https://openreview.net/forum?id=ONwL9ucoYG&noteId=WfiCNkAi1u
> > >
> > > (3) RoCLIP considerably outperforms defense techniques proposed for fine tuning:
> > > https://openreview.net/forum?id=ONwL9ucoYG&noteId=sLv3whlUdj
> > >
> > > Considering the remarkable performance of RoCLIP and the ever-increasing application of the multimodal models, we hope that the reviewer considers further increasing their score to support our work. Please let us know if we can provide any additional results! We appreciate your valuable feedback that helped us improve our submission.

---

### Official Review · Reviewer_8GQL · 2023-06-28

**Soundness:** 2 fair
**Presentation:** 3 good
**Contribution:** 3 good
**Rating:** 7
**Confidence:** 4

**Summary:**

This paper introduces ROCLIP, the first robust method for pre-training multimodal vision-language models that effectively combats targeted data poisoning and backdoor attacks. These attacks pose a significant threat to large multimodal models like CLIP, which learn from millions of image-caption pairs from the internet, making them particularly vulnerable. To tackle this, ROCLIP disassociates poisoned image-caption pairs by using a large, variable pool of random captions and matches each image with the most similar text in the pool, rather than its own caption. The paper demonstrates through comprehensive experiments that ROCLIP significantly reduces the success rate of targeted data poisoning and backdoor attacks during the pre-training of CLIP, and enhances the model's linear probe performance by 10%, while maintaining similar zero-shot performance.

**Strengths:**

1. The idea is novel and simple.

2. The paper is well-written and easy to follow.

3. The experiments show that ROCLIP can effectively defend against poisoning attacks and backdoor attacks.

**Weaknesses:**

1.  Some details are not clearly introduced. For example, the author should explicitly define the threat model, which will make the paper more clear.

2. Lack of adaptive defense discussion. If the adversary is aware that ROCLIP has been adopted, can the adversary adaptively bypass the defense? For example, as shown in the right part of Fig. 3, although the target image is no longer matching to the poisoned captions, the most similar NN caption doesn't match the image content. So, if the adversary can utilize such a mismatch, can ROCLIP be adaptively bypassed?

3. Only a low poison ratio setting is evaluated. In Carlini's work, it shows that a low poison ratio can achieve a good attack performance. However, as a defense work, the author should discuss the effectiveness of ROCLIP under different poison ratios.

4. The explanation for Eq. 1 is not clear. If my understanding is correct, this denotes the symmetric loss function. However, it makes the readers a little confused. The two parts of Eq. 1 cannot show their difference. The authors should explain their difference more clearly.

**Questions:**

Can ROCLIP defend against adaptive attacks?

Why does ROCLIP achieve a lower accuracy in a zero-shot setting but a higher accuracy in a linear-probe setting?

What's the performance of ROCLIP under different poison ratios?

Why does the larger pool size have a lower time cost? Intuitively, fetching NN caption from a larger pool will cause higher time costs.

**Limitations:**

Experiments are not thorough enough. Some details are not clearly introduced.

---

> ### Author Rebuttal · Authors · 2023-08-10
>
> We thank the reviewers for recognizing the novelty, effectiveness, and simplicity of our method and the clarity in our writing.
>
> (1) For defining the threat model, we define our threat model as follow:
>
> Adversary Objective: The primary objective of the adversary is to manipulate the output representations of the multimodal contrastive model, such that  certain images are misclassified into desired categories instead of their true categories, while the other images are classified correctly.
>
> Adversary Capabilities: We assume that the adversary has limited control over the training data, and can inject a small number (< 0.002% of the total dataset) of examples into the training dataset. Adversary also has knowledge of the model structure, the training algorithm, and the hyperparameter used by their victim, but the attacker cannot modify the training process directly.
>
> We will add this discussion to our revised version.
>
> (2) Our method leverages the observation that early in training, poisoned image-caption pairs are farther away in the representation space compared to clean pairs. Based on this, there are two potential adaptive attacks that can be crafted if adversity knows that RoCLIP is applied for pretraining:
>
> a. For both targeted data poisoning and backdoor attacks, one way to circumvent our defense is to simply increase the poison ratio. In this way, the poisoned image and caption pairs will get closer to each other earlier in training. Since RoCLIP matches an image to the caption closest to the image representation in the pool, under a higher poison ratio the poisoned image and captions may still be matched together.
>
> b.The other adaptive attack is to use adversarial captions that are semantically more similar (but do not correspond) to a target image. For example, a dog image paired with captions related to cats will be more difficult to defend. Since dog and cat-related captions have a higher similarity in the text representation space, there is a higher chance that the dog image is matched to a cat caption (as its nearest neighbor in the pool). However, this approach substantially limits the choice of attackers’ target images as well as adversarial captions.
>
> To illustrate our method’s effectiveness in these scenarios, we conduct the following experiments in 1M data setting:
>
> a. We increased the adversarial captions to {100, 256, 512}. Note that this  targeted poison ratio (0.000512) is higher than the max poison ratio (0.00017) considered in Carlini et al. As shown from the table below, our method is able to defend the model. Note that, to defend more poisons, we used RoCLIP with an increased K=2 (vs K=3 in the paper), where K is the frequency of applying the NN loss (every K epoch) . This results in a stronger defense against attacks at the expense of slightly compromising some of the downstream performances. Specifically, there is a drop in the zero-shot performance on CIFAR10, but the linear-probe performance and the zero-shot performances on other datasets are still on par with CLIP.
>
> | Model  | # poison    | PSR      | LP C10 | LP C100 | ZS C10 | ZS C100 |
> |--------|-------------|----------|--------|---------|--------|---------|
> | RoCLIP | 100/256/512 | 0%/0%/0% | 69.7%  | 46.6%   | 21.3%  | 7.27%   |
> | CLIP   | -           | -        | 70.5%  | 45.8%   | 30.14% | 7.29%   |
>
>
> b. We include 8 pairs of poisoned image-captions whose categories are visually similar, like basketball with football-related captions; dog with cat-related captions, etc.
>
> As shown from the table, our method is able to defend the attacks effectively, bringing the targeted poison success rate (PSR) down to 25%, despite the poison being more challenging than random poisons.
>
> | model  | # poison | # pairs | PSR |
> |--------|----------|-----|-----|
> | RoCLIP | 15       | 8| 25% |
> | CLIP   | 15       | 8| 75% |
>
> (3) As reported above, our model is able to defend against attacks with higher targeted poison ratio up to 0.000512. Note that 0.00017 is the highest targeted poison ratio applied in [1]’s setting. However, this may be too high for a realistic real-world scenario.
>
> (4)  For the explanation of Eq. 1:
>
> The first part of the equation denotes contrasting the image with the captions. The negative pairs are the fixed image $z^I_j$ and all the captions ${z^T_k\}_{k=1}^N$ in the mini-batch corresponding to the other images. The second part of the equation denotes contrasting captions with the images. The negative pairs are the fixed caption, $z^T_k$ and all the images $\{z^I_j\}$  in the mini-batch corresponding to the other captions.
>
> Q1&3: See (2)
>
> Q2:  For the RoCLIP having lower accuracy in zero-shot settings and higher accuracy in the linear probe settings:
>
> The improvement in linear-probe settings are due to the data augmentation used by RoCLIP during training. In table 5 (explained in line 332), we have an ablation study isolating the effect of pool and augmentation on the performance of RoCLIP. We see that data augmentation alone considerably improves the linear probe accuracy.
>
> The small performance drops in zero-shot settings are due to the nearest-neighbor matching in the pool used by RoCLIP: While this is essential to break the attacks, some of the images may not be accurately matched to their corresponding categories, and instead are matched to neighboring categories with similar representations.
>
> Q4: For larger pool size having a lower time cost:
> We re-measured the per-epoch run time for our method in 100K settings with different pool sizes. The result is shown in the table below. The overhead is mostly caused by augmentation and finding the NNs has a negligible overhead, thus pool of different size has a similar per-epoch run time.
>
> | model               | time(s) |
> |---------------------|---------|
> | CLIP                | 506.02  |
> | CLIP w/augmentation | 536.96  |
> | RoCLIP (pool=256)   | 538.31  |
> | RoCLIP (pool=2048)  | 538.69  |
> | RoCLIP (pool=21845) | 539.47  |

---

> > ### Comment · Reviewer_8GQL · 2023-08-14
> > **Response to Author's Rebuttal**
> >
> > The rebuttal is satisfactory. Most of my concerns are addressed. However, I encourage the authors to explore a higher poisoning ratio (e.g., 0.1 - 1%), which makes more sense for a defense paper.

---

> > > ### Author Response · Authors · 2023-08-18
> > >
> > > We thank the reviewer for reading our rebuttal and to hear that we could address most concerns.
> > >
> > > Please note that a much lower poison rate is enough to successfully poison multimodal models (0.0001% for multimodal vs 1% for supervised learning), and hence this poison rate is considered in [1], [2], and our original submission. Considering the very large size of the pretraining data for multimodal models, this small poison rate translates to a considerable “poison number” that needs to be generated by the adversary. Nevertheless, based on the reviewer’s suggestion, we conducted new experiments with 0.5% and 1% poison ratio. The table below shows RoCLIP’s performance on 1M data with these higher poisoning ratios. As discussed in our rebuttal, increasing the frequency of RoCLIP loss to K=2 (applying the RoCLIP loss every 2 epochs), our method can successfully defend against a much higher poison rate. The table below shows that **under 0.5%,1% poison ratio, RoCLIP remarkably drops the attack success rate from 100% to 0% and 12.5%, respectively!**  Note that the poisoning ratio does not change the performance of the model trained with RoCLIP, and zero-shot and linear-probe are the same as what we reported in our rebuttal.
> > >
> > > | Model  | # poison      | poisoning ratio | PSR   |
> > > |--------|---------------|-----------------|-------|
> > > | RoCLIP | 5,000         | **0.5%**            | **0%**    |
> > > | RoCLIP | 10,000        | **1.0%**             | **12.5%** |
> > > | CLIP   | 5,000(10,000) | **0.5%(1.0%)**      | **100%**  |
> > >
> > >
> > > We hope that our additional experiments could address the remaining concerns and you consider increasing your score. We appreciate your valuable feedback on improving our submission.
> > >
> > >
> > > [1] Yang, Z., He, X., Li, Z., Backes, M., Humbert, M., Berrang, P. and Zhang, Y., 2022. Data Poisoning Attacks Against Multimodal Encoders. arXiv preprint arXiv:2209.15266.
> > >
> > > [2]Carlini, N. and Terzis, A., 2021, October. Poisoning and Backdooring Contrastive Learning. In International Conference on Learning Representations.

---

> > > > ### Comment · Reviewer_8GQL · 2023-08-18
> > > > **Response to Author's Rebuttal**
> > > >
> > > > Thank you for the author's responses.  Indeed, a low poisoning ratio for an attack implies a robust and practical attacker, as corroborated by the papers you cited. However, when considering defense mechanisms, it is essential to assume a more potent adversary that employs a higher poisoning ratio. This not only aligns with the realistic challenges faced in the field but also aids the community in understanding the effective boundaries of your work. The experimental results have thoroughly addressed my concerns. Therefore, I have revised my rating to a 7.

---

> > > > > ### Author Response · Authors · 2023-08-18
> > > > >
> > > > > We sincerely thank the reviewer for the insightful comments, which helped us improve our work. We will incorporate this discussion regarding the higher poisoning ratio and our new results in the revised version of our paper. Many thanks for your continued support!

---

### Official Review · Reviewer_cYJD · 2023-07-06

**Soundness:** 3 good
**Presentation:** 3 good
**Contribution:** 3 good
**Rating:** 6
**Confidence:** 3

**Summary:**

This paper proposes ROCLIP for effective and roubust pre-training multimodal vision-language models against both targeted data poisoning and backdoor attacks.
ROCLIP breaks the associationbetween poisoned image-caption pairs, instead, it uses image-augmented_caption pairs to remove backdoors.
ROCLIP significantly decreases the attack success rate, and effectively improves the model’s linear probe performance.

**Strengths:**

1.The paper is well-written and easy to follow.

2.The proposed idea is simple yet effective.

3.The defense on multimodality is a relative new area, and there must be a lot of practical value.

**Weaknesses:**

1.In Table 3, the performance of “Clean CLIP” is better than the performance of “RoCLIP”? Can the author illustrate the advantages of “RoCLIP” compared to “Clean CLIP”?

2.For the defense baselines, the paper only compared with “Clean CLIP”, are there other defense baselines? (I know reference [1] also proposes a defense method, but it is recently published on ICLM’23. so it might be reasonable that this paper does not compared with [1].)

3.There is totally two attack baselines (Targeted image attacks and backdoor attacks) in the experiments. Are there other backdoor attack baselines?
The CleanCLIP[3] uses several other backdoor attack methods. Is it possible to extend your RoCLIP defense against other attacks?

4.Writting Suggestions:

1)The annotation is not consistent in the paper. For example, “tab.x”, “Tab.x”, “Table x” should be consistent.

2)“image-caption pairs” and “image-captions pairs” appear several times. Should all of them be “image-caption pairs”?



[1] Yang, Z., He, X., Li, Z., Backes, M., Humbert, M., Berrang, P. and Zhang, Y., 2022. Data Poisoning Attacks Against Multimodal Encoders. arXiv preprint arXiv:2209.15266.

[2]Carlini, N. and Terzis, A., 2021, October. Poisoning and Backdooring Contrastive Learning. In International Conference on Learning Representations.

[3] Bansal, H., Singhi, N., Yang, Y., Yin, F., Grover, A. and Chang, K.W., 2023. CleanCLIP: Mitigating Data Poisoning Attacks in Multimodal Contrastive Learning. arXiv preprint arXiv:2303.03323.

**Questions:**

1.In experiments, how many backdoored models are using? Normally we need more than one backdoored model. The paper does not mention.

**Limitations:**

The authors adequately addressed the limitations.

---

> ### Author Rebuttal · Authors · 2023-08-10
>
> We thank the reviewer for recognizing the significance of our problem, the importance of defending multimodal models, the effectiveness and simplicity of our method, as well as the clarity in our writing.
>
> (1) Difference between “Clean CLIP” and “RoCLIP”:
>
> In Table 3, we used the notation “Clean CLIP” as opposed to “Poisoned CLIP” to refer to  the CLIP model trained on clean data. This is not the CleanCLIP method of [3]. We apologize for the confusion and will change the name of the corresponding column in the table.
>
> (2) Additional baselines:
>
> RoCLIP is the first defense against targeted data poisoning and backdoor attacks on CLIP during pre-training. Both methods proposed in [1] and [3] are applicable to fine-tuning and are not effective for pre-training.
>
> The methods proposed in [1] targets data poisoning during fine-tuning and requires a clean pre-trained CLIP model. Specifically, they proposed two methods:
>
> The first strategy of [1] relies on a CLIP model pre-trained on clean data to calculate the cosine similarity between all the image-caption pairs and filters out potential poisoned pairs that have a smaller cosine similarity than a pre-specified  threshold. However, this approach is not applicable to pre-training as it requires a CLIP model pre-trained on clean data, which is not available during pre-training. In addition, an appropriate threshold depends on the image and captions of the targeted attacks and varies for different sizes of backdoors. A large threshold drops the model performance and a small threshold cannot break the attack. While fine-tuning is less sensitive (as the model changes less), the performance of pre-training is highly affected by such a threshold. Hence, this method is not appropriate for pre-training.
>
> The second strategy of [1] is fine-tuning the model on an additional clean dataset of comparable size to the pre-training dataset. This is clearly not applicable to pre-training.
>
> The method of [3] aims at eliminating backdoors from a poisoned CLIP during fine-tuning. To do so, it fine-tunes the model in an unsupervised manner on a clean subset of the pre-trained dataset. This method is also clearly not applicable to pre-training.
>
> Defending CLIP against targeted data poisoning and backdoor attacks during pre-training is more difficult than fine-tuning and demands more intricate techniques. Our method is the first successful attempt towards addressing this problem.
>
> (3) For additional backdoor attack methods:
>
> We have conducted new experiments with invisible triggers proposed in [2] and [3]. The results are shown below. We conducted our experiments on 100K data and used 90 pairs of backdoored image-caption pairs for Blended triggers and 256 pairs of backdoored image-caption pairs for WaNet triggers. Note that our backdoor attack ratios (0.0009 for Blended, 0.00256 for WaNet) are higher than the ones used by CleanCLIP (0.0005 for Blended and WaNet).
>
> As shown in the table, our method is able to defend against various backdoor attacks and bring the attack success rate down to nearly 0%.
> |        | Blended | WaNet |
> |--------|---------|-------|
> | CLIP   | 28%     | 43.4% |
> | RoCLIP | 0.33%   | 0.67% |
>
> (4) For writing suggestions:
>
> We thank the reviewer for bringing this to our attention. We will fix the typos in our revised version and make the writing consistent.
>
> (5) For how many backdoored models did we use:
>
> In our experiment, we used 4 backdoored models. We will specify this in our revised version.

---

> > ### Comment · Reviewer_cYJD · 2023-08-14
> >
> > Thanks for the responses. You have released some of my concerns. However, I am still not sure about 'Weakness 2 & 3'.

---

> > > ### Author Response · Authors · 2023-08-18
> > > **Weakness 2 [1]**
> > >
> > > We thank the reviewer for reading our rebuttal and we’re glad that we could address the majority of the questions.
> > >
> > > For weakness 2:
> > >
> > > While we are confident that [1], [3] are the only existing methods for defending multimodal models against targeted data poisoning and backdoor attacks, as we discussed in the paper and our rebuttal, they are originally proposed to defend the model during fine-tuning and not pretraining (as we consider in our paper). Here, we provide additional discussion and report new experimental results for applying such methods during pretraining.
> > >
> > > **Weakness 2 [1]:**
> > >
> > > [1] proposed **using a pretrained** model to calculate and compare cosine similarity between poisoned image-caption pairs and clean image-caption pairs. Then, it finds a threshold to separate the majority of poisoned pairs from clean pairs, and discard all the pairs with lower similarity than the threshold.
> > >
> > > We applied the above idea to pretraining, in a **cheating experiment**. That is, we consider the poisoning ratios of 0.005%~0.5%, and calculate the cosine similarity between the **known generated poisoned image-caption pairs** (note that in practice the poisons are not known) at epoch 1. We chose epoch 1, as the separation between cosine similarity of poisoned and clean image caption paris is largest at epoch 1 (see Fig 2). Then we set the threshold such that a certain fraction of the poisoned pairs are discarded, and filter examples with cosine similarity lower than this threshold.
> > > The following table shows the fraction of clean pairs that are discarded using different thresholds. We see that regardless of the threshold, **a very large fraction of clean pairs are discarded**.
> > >
> > >
> > > | poison-discard %/poisoning% | 0.005% | 0.01% | 0.1%   | 0.5%  |
> > > |----------------------------|--------|-------|--------|-------|
> > > | 75%                        | 15.7% (*) | 44.3% | 98.3%  | 95.8% |
> > > | 85%                        | 23.3% (*) | 62.3% | 98.9%  | 96.3% |
> > > | 95%                        | 38.1%  | 87.7% | 99.55% | 97.1% |
> > >
> > >
> > > We trained and evaluated the two models when less than 30% of the clean data was discarded, marked with (*), and both models were still poisoned. With 75% and 85% of the poison discarded, the targeted data poisoning attacks still have 53.84% poison success rates. This confirms the ineffectiveness of the above method during pretraining. On the other hand, RoCLIP is able to drop the success rate to 0% (using K=2) and 12.5% (using K=3), and provides a much better trade-off between the attack success rate and model’s performance and **can defend a much higher poison number up to 1%** (note that for a particular K, performance of RoCLIP is not affected by the poison number).
> > >
> > >
> > > |  Model      | K | Poison % | Poison Discard % | PSR    | LP C10 | LP C100 | ZS C10 | ZS C100 |
> > > |--------|---|----------|------------------|--------|--------|---------|--------|---------|
> > > | CLIP   | - | 0.005%   | -                | **93.75%**| 70.5%  | 45.83%  | 34.95% | 7.29%   |
> > > | CLIP+threshold (*)   | - | 0.005%   | **75%**              | 58.34% | 70.25% | 45.62%  | 27.92% | 7.34%   |
> > > | CLIP+threshold (*)   | - | 0.005%   | **85%**              | 58.34% | 69.9%  | 45.59%  | 24.05% | 6.51%   |
> > > | RoCLIP | 3 | 0.005%   | -                | **12.5%**  | 78.99% | 57.82%  | 30.14% | 9.52%   |
> > > | RoCLIP | 2 | 0.005%   | -                | **0%**     | 69.7%  | 46.6%   | 21.3%  | 7.27%   |
> > > | RoCLIP | 2 | **0.1%**     | -                | **0%**     | 69.7%  | 46.6%   | 21.3%  | 7.27%   |
> > > | RoCLIP | 2 | **0.5%**     | -               | **0%**     | 69.7%  | 46.6%   | 21.3%  | 7.27%   |
> > > | RoCLIP | 2 | **1%**     | -               | **12.5%**     | 69.7%  | 46.6%   | 21.3%  | 7.27%   |
> > >
> > >
> > > We note that, while the lower “average” similarity of poisoned pairs during the initial training epochs is an important factor for the success of RoCLIP, **this is not the only factor**. The other key factor for the success of RoCLIP is that **poisoned images and captions are not close to the other images and captions in the same category**. For example, a poisoned dog image that is captioned adversarially as a deer, remains far away from the rest of dog images in the representation space, during the initial training epochs (this is discussed in lines 171-173 of the paper). Hence, nearest-neighbor matching done by RoCLIP can effectively break the association of poisoned image caption pairs.

---

> > > ### Author Response · Authors · 2023-08-18
> > > **Weakness 2 [1, 3] & Weakness 3**
> > >
> > > **Weakness 2 [1, 3]**:
> > >
> > > [1] proposed fine-tuning on a clean dataset (of the same size as pretraining) with CLIP loss, and [3] proposed fine-tuning on a clean subset of size 100K of the original training dataset (same size as the pretraining data we use in this experiment) with CLIP loss+in-modality contrastive loss on both image and text modalities. Since a clean dataset is not available during pretraining, here we use the CleanCLIP loss (CLIP loss + in-modality contrastive loss) [3] to pretrain CLIP on the (poisoned) data.
> > >
> > > We conduct our experiment on 100K data, with 150 pairs of backdoored image-captions pairs and 15 targeted poisoned pairs. The following table shows that while the in-modality loss enables CleanCLIP to achieve a good performance, CleanCLIP cannot successfully defend against targeted data poisoning and backdoor attacks during pretraining. On the other hand, RoCLIP effectively drops the attack success rate to 0% for both targeted poisoning and backdoor attacks. This further confirms the effectiveness of RoCLIP.
> > > | Model     | # poison | # backdoor | PSR | BDR | LP C10 | LP C100 |
> > > |-----------|----------|------------|-----|-----|--------|---------|
> > > | RoCLIP    | 15       | 150        | **0%**  | **0%**  | 67.07% | 39.64%  |
> > > | CleanCLIP | 15       | 150        | **25%** | **49%** | 67.75% | 43.22%  |
> > > | CLIP | 15       | 150        | **75%** | **49%** | 62.09% | 37.41%  |
> > >
> > >
> > > **For weakness 3**:
> > >
> > > Our original submission and rebuttal covered all (except one) of targeted poisoning and backdoor attacks considered by prior work in the multimodal setting (see [1,2,3]). We conducted new experiments on the only remaining backdoor attack in [3], namely “label-consistent backdoor attack” to our attack baseline as well. Consistent to our rebuttal setting, we conducted our experiments with 512 label-consistent backdoors in the 100K data setting. We see that RoCLIP is able to defend the attack successfully. Note that the type of the backdoor doesn’t change the performance of RoCLIP, and the zero-shot and linear probe are the same as what we reported in our rebuttal.
> > >
> > > | Model  | # backdoor | BDR   |
> > > |--------|------------|-------|
> > > | RoCLIP | 512        | **0%**    |
> > > | CLIP   | 512        | **97.3%** |
> > >
> > >
> > > Note that the targeted class attack in [1], which targeted a specific class of images instead of a single image is only applicable to fine-tuning, where classes are well-defined (e.g. [1] applied this to CoCo dataset). However, this attack is not applicable to pretraining since the notion of class/label is not defined (data only contains images and captions).
> > >
> > >
> > > We hope that our additional experiments could address the remaining concerns and you consider increasing your score. We appreciate your valuable feedback on improving our submission.

---

### Official Review · Reviewer_99eL · 2023-07-07

**Soundness:** 3 good
**Presentation:** 3 good
**Contribution:** 3 good
**Rating:** 5
**Confidence:** 4

**Summary:**

The paper introduces ROCLIP, a method for robust pre-training of multimodal vision-language models, such as CLIP, against targeted data poisoning and backdoor attacks. ROCLIP addresses the vulnerability by breaking the association between poisoned image-caption pairs. It does this by considering a large and varying pool of random captions and matching every image with the text that is most similar to it in the pool, rather than its original caption. The method also employs image and text augmentations to further enhance the defense and improve model performance. Extensive experiments show that ROCLIP significantly reduces the success rate of poison and backdoor attacks during CLIP pre-training, while improving the model's linear probe performance and maintaining similar zero-shot performance compared to CLIP.

**Strengths:**

1.	ROCLIP introduces a novel approach for robust pre-training of multimodal vision-language models against backdoor attacks, which has been a largely unaddressed problem.
2.	The method uses a simple yet effective strategy of breaking the association between poisoned image-caption pairs by matching every image with the text most similar to it in a pool of random captions.
3.	Extensive experiments show promising results, where ROCLIP not only decreases the success rates of poison and backdoor attacks significantly but also enhances the model's linear probe performance without compromising the zero-shot performance.

**Weaknesses:**

1.	While ROCLIP shows promising results against targeted data poisoning and backdoor attacks, it is uncertain how it would fare against more subtle or sophisticated attacks or those that adapt to its defenses. A valuable extension of this work would be to evaluate the robustness of ROCLIP against varying poison ratios in the training data to assess its effectiveness under more intense or evolved attack scenarios.

**Questions:**

1.	Could you provide some potential adaptive attack for the proposed method?
2.	The effectiveness of ROCLIP in the context of smaller datasets, such as those in the medical domain, remains unclear. Given the specialized and often limited size of such datasets, it would be insightful to evaluate how ROCLIP performs under these circumstances. This could better illustrate the versatility and adaptability of the method across different domains and data volumes.

**Limitations:**

This paper has no negative societal impact.

---

> ### Author Rebuttal · Authors · 2023-08-10
>
> We thank the reviewer for acknowledging the importance of this unaddressed problem for multimodal models, the extensiveness of our experiments, as well as the novelty and effectiveness of our method against targeted data poisoning and backdoor attacks.
>
>
> (1) Our method leverages the observation that early in training, poisoned image-caption pairs are farther away in the representation space compared to clean pairs. Based on this, there are two potential adaptive attacks that can be crafted if adversity knows that RoCLIP is applied for pretraining:
>
>
> a. For both targeted data poisoning and backdoor attacks, one way to circumvent our defense is to simply increase the poison ratio. In this way, the poisoned image and caption pairs will get closer to each other earlier in training. Since RoCLIP matches an image to the caption closest to the image representation in the pool, under a higher poison ratio the poisoned image and captions may still be matched together.
>
>
> b.The other adaptive attack is to use adversarial captions that are semantically more similar (but do not correspond) to a target image. For example, a dog image paired with captions related to cats will be more difficult to defend. Since dog and cat-related captions have a higher similarity in the text representation space, there is a higher chance that the dog image is matched to a cat caption (as its nearest neighbor in the pool). However, this approach substantially limits the choice of attackers’ target images as well as adversarial captions.
>
>
> To illustrate our method’s effectiveness in these scenarios, we conduct the following experiments in 1M data setting:
>
>
> 1. We increased the adversarial captions to {100, 256, 512}. Note that this  targeted poison ratio (0.000512) is higher than the max poison ratio (0.00017) considered in Carlini et al. As shown from the table below, our method is able to defend the model. Note that, to defend more poisons, we used RoCLIP with an increased K=2 (vs K=3 in the paper), where K is the frequency of applying the NN loss (per K epoch). This results in a stronger defense against attacks at the expense of slightly compromising some of the downstream performances. Specifically, there is a drop in the zero-shot performance on CIFAR10, but the linear-probe performance and the zero-shot performances on other datasets are still on par with CLIP.
>
>
> | Model  | # poison    | PSR      | LP C10 | LP C100 | ZS C10 | ZS C100 |
> |--------|-------------|----------|--------|---------|--------|---------|
> | RoCLIP | 100/256/512 | 0%/0%/0% | 69.7%  | 46.6%   | 21.3%  | 7.27%   |
> | CLIP   | -           | -        | 70.5%  | 45.8%   | 30.14% | 7.29%   |
>
>
> 2. We include 8 pairs of poisoned image-captions whose categories are visually similar. The categories include:
> basketball image with football-related captions; dog with cat-related captions; bee with butterfly-related captions, etc. We generated 15 adversarial captions for each poisoned image.
>
>
> As shown from the table, our method is able to defend the attacks effectively, bringing the poison success rate (PSR) down to 25%, despite the poison being more challenging than random poisons.
>
>
> | model  | # poison | # pairs | PSR |
> |--------|----------|-----|-----|
> | RoCLIP | 15       | 8| 25% |
> | CLIP   | 15       | 8| 75% |
>
>
> (2) For smaller and more specialized datasets:
>
>
> Note that, in our ablation studies, we included experiments with 100K data, a much smaller data size compared to our standard experiments. As shown in section 6.1 and section 6.2 (line 315~ line 341), when using 100K data, our method is able to defend against targeted data poisoning attacks with 0.015% poison ratio, higher than the 0.005% used in the 1M setting, and achieve 0% poison success rate. The summary of the results are shown in table 4 and table 5.
>
>
> To further illustrate our method’s effectiveness in smaller datasets, we train on a smaller subset of CC3M, of size 50K. We used a higher targeted poison ratio (0.0003) compared to our 100K experiment (0.00015), by generating 15 adversarial captions for each image. As shown in the table below, our method is able to defend the attacks effectively, bringing the poison success rate (PSR) down to 0%.
>
>
> | model  | # poison | # pairs | PSR   |
> |--------|----------|---------|-------|
> | RoCLIP | 15       | 8       | 0%    |
> | CLIP   | 15       | 8       | 87.5% |

---

> > ### Comment · Reviewer_99eL · 2023-08-15
> > **Response to Rebuttal**
> >
> > I appreciate the authors' responses to my comments. Your responses have alleviated some of my concerns, and I have decided to maintain my original rating.

---

> > > ### Author Response · Authors · 2023-08-18
> > >
> > > We thank the reviewer for their response and we are glad that we could resolve most of the concerns. While we are not sure which concern is left unaddressed, we provide the following new experimental evaluations, confirming that:
> > >
> > > (1) **RoCLIP can effectively defend much higher poison ratios (0.5-1%)**: https://openreview.net/forum?id=ONwL9ucoYG&noteId=WYaDLJFW0j
> > >
> > > (2) **RoCLIP can successfully defend a new backdoor attack**: https://openreview.net/forum?id=ONwL9ucoYG&noteId=WfiCNkAi1u
> > >
> > > (3) **RoCLIP considerably outperforms defense techniques proposed for fine tuning**:https://openreview.net/forum?id=ONwL9ucoYG&noteId=sLv3whlUdj
> > >
> > > Regarding pretraining on smaller datasets, we provided experiments on 50K and 100K examples in our paper and rebuttal to confirm the effectiveness of RoCLIP. For specialized datasets, a pretrained CLIP is often fine-tuned on smaller specialized data, for which [1,3] proposed defense techniques. In contrast, our method focuses mainly on the pretraining phase. Nevertheless, we tried finding a specialized image-caption datasets that is suitable for multimodal learning, but unfortunately we could not find any dataset that can be used for training CLIP. If the reviewer can refer us to some datasets good for pretraining, we will conduct new experiments on this dataset.
> > >
> > > We hope that our additional experiments could address the remaining concerns. Considering the remarkable performance of RoCLIP and the ever-increasing application of the multimodal models, we hope that the reviewer considers increasing their score to support our work. Please let us know if we can provide any additional results! We appreciate your valuable feedback that helped us improve our submission.
> > >
> > > [1] Yang, Z., He, X., Li, Z., Backes, M., Humbert, M., Berrang, P. and Zhang, Y., 2022. Data Poisoning Attacks Against Multimodal Encoders. arXiv preprint arXiv:2209.15266.
> > >
> > > [2]Carlini, N. and Terzis, A., 2021, October. Poisoning and Backdooring Contrastive Learning. In International Conference on Learning Representations.
> > >
> > > [3] Bansal, H., Singhi, N., Yang, Y., Yin, F., Grover, A. and Chang, K.W., 2023. CleanCLIP: Mitigating Data Poisoning Attacks in Multimodal Contrastive Learning. arXiv preprint arXiv:2303.03323.

---

### Decision · Program_Chairs · 2023-09-21

**Decision:**

Accept (poster)

**Comment:**

In this paper, the authors propose ROCLIP that is claimed to be the first effective method for robust pre-training multimodal vision-language models against targeted data poisoning and backdoor attacks. After the rebuttal process, the responses from the authors mostly satisfy the reviewers and the reviewers reach a consistent consensus of raising the scores and accepting this manuscript. Taking all positive comments from the reviewers into consideration, this paper is recommended to be accepted!